# Characteristics of Carbon Nanotube Cold Cathode Triode Electron Gun Driven by MOSFET Working at Subthreshold Region

**DOI:** 10.3390/nano14151260

**Published:** 2024-07-28

**Authors:** Yajie Guo, Baohong Li, Yu Zhang, Shaozhi Deng, Jun Chen

**Affiliations:** State Key Laboratory of Optoelectronic Materials and Technologies, Guangdong Province Key Laboratory of Display Material and Technology, School of Electronics and Information Technology, Sun Yat-sen University, Guangzhou 510275, China; guoyj37@mail2.sysu.edu.cn (Y.G.); lbaoh@mail.sysu.edu.cn (B.L.); stszhyu@mail.sysu.edu.cn (Y.Z.); stsdsz@mail.sysu.edu.cn (S.D.)

**Keywords:** carbon nanotube, field emission, cold cathode, MOSFET, electron gun

## Abstract

The carbon nanotube cold cathode has important applications in the X-ray source, microwave tube, neutralizer, etc. In this study, the characteristics of carbon nanotube (CNT) electron gun in series with metal-oxide-semiconductor field-effect transistor (MOSFET) were studied. CNTs were prepared on a stainless steel substrate by chemical vapor deposition and assembled with a mesh gate to form an electron gun. The anode current of the electron gun can be accurately regulated by precisely controlling the MOSFET gate voltage in the subthreshold region from 1 to 40 µA. The current stability measurements show the cathode current fluctuation was 0.87% under 10 h continuous operation, and the corresponding anode current fluctuation was 2.3%. The result has demonstrated that the MOSFET can be applied for the precise control of the CNT electron gun and greatly improve current stability.

## 1. Introduction

Since Sumio Iijima discovered carbon nanotubes (CNTs) in 1991, their applications in electronic and optoelectronic devices have been extensively explored [1,2,3,4]. One of the important applications of CNTs is field emission cold cathode for vacuum microelectronic devices. As a one-dimensional material, CNTs are regarded as ideal cold cathode materials because of their excellent field emission performance due to their high aspect ratio, high electron conductivity, and high thermal conductivity. CNT cold cathodes demonstrate a longer operational lifetime, faster response speed, and lower power consumption. Furthermore, CNTs can be prepared using a low-cost self-assembly method and overcome the shortcomings of a complicated fabrication process for Spindt tip arrays [5,6]. CNTs have, thus, been widely studied for potential applications for field emission displays (FEDs), X-ray sources, microwave power devices, electron microscopes, and other vacuum microelectronic devices [7,8,9,10,11,12,13,14].

A cold cathode electron gun is the key component of many vacuum microelectronic devices [15,16,17]. A cold cathode electron gun typically has a triode structure consisting of a cathode, gate, and anode. In the discrete triode device, the gate usually is a mesh mechanically installed above the cathode. By applying voltage, the gate controls the emission of electrons from the cathode and the anode collects the emitted electrons. Carbon nanotube cold cathode triode electron guns have been employed in X-ray and microwave sources [18,19,20,21,22].

However, the gate voltage of cold cathode electron guns remains relatively high because of the large gate-to-cathode distance if a discrete triode structure is used [23,24,25]. On the other hand, the field emission (FE) current follows the Fowler–Nordheim (F-N) equation, which has an exponential relationship with voltage [26]. Therefore, it is difficult for the gate to precisely modulate the emission current. Moreover, the current instability in the field emission current is inevitable due to the complicated processes including surface atom migration, residue gas molecular adsorption and desorption, ion bombardment, etc. [27,28,29], while in some applications such as cold cathode X-ray sources for intraoperative radiotherapy (IORT) and neutralizers for ion engines, precise emission current control and high stable emission are necessary [30,31].

The above-mentioned issue limits the use of CNT triode structure electron guns in those applications with high-precision control demand. To solve the problems, controlling the electron gun with an extra active control component such as a thin-film transistor (TFT) or metal-oxide-semiconductor field-effect transistor (MOSFET) is an effective approach.

Triode structure electron sources using an integrated control device have been reported, in which the cold cathode is prepared directly with TFT on MOSFET using a microfabrication process. Cho et al. integrated amorphous silicon thin-film transistors (a-Si TFT) with Mo tip arrays and the emission currents of the fabricated cathode can be effectively controlled by the a-Si TFT gate voltage as low as 25 V [32]. Yang et al. have reported a MOSFET-controlled ZnO nanowire (NW) cathode. ZnO nanowires (NWs) were grown on the drain of a MOSFET to form an on-chip integrated device by using a low-temperature solution-phase method. The emission current can be precisely tuned from 0.2 nA to 1.15 μA by modulating the MOSFET gate voltage from 0.8385 V to 1.5255 V. The field emission current stability of the MOSFET-controlled ZnO NWs was also increased remarkably [33]. Cheng et al. have developed a device scheme of CNTs incorporating a TFT. CNTs are directly integrated in the drain region of the TFT and the emission current fluctuation of the TFT-controlled carbon nanotubes can be reduced to within 2%, which has great potential for future applications in field emission display and vacuum microelectronics [34].

For discrete CNT cold cathode devices, it is difficult to directly prepare the active control devices on the electron gun structure. Therefore, researchers usually connect the CNT cold cathode in series with the active control devices, such as MOSFET. For example, Bae et al. reported MOSFET-controlled diode CNT emitters and showed excellent field emission current stability. The ratio of change in the current of CNT emitters with an externally connected MOSFET was decreased to 0.45% from 9.53% for that without the MOSFET for over 100 h continuous operation [35]. Deng et al. used transistors to regulate the emission uniformity of the lighting element pixels of a field emission display [36]. Guo et al. have studied a gated CNT cold cathode electron gun with an insulated gate bipolar transistor (IGBT) modulation and highly stable cathode current with less than 0.5% fluctuation was achieved for 50 min continuous operation. Their results also demonstrate IGBT modulation is an effective way to achieve high current stability in a gated carbon nanotube cold cathode X-ray source [37].

In some reports, a more complicated circuit was used for controlling the CNT electron gun. Lee et al. stabilized field emission current from a gated CNT electron gun by a cascade active-current-control (ACC) circuit consisting of two high-voltage field-effect transistors (HV-FETs). The field emission current had shown a standard deviation of less than 0.4%. The cascade ACC also had the potential to equalize the dose of X-ray tube, thereby improving image quality for stationary digital breast tomosynthesis (sDBT) and inverse-geometry computed tomography (IGCT) [38].

In this study, the characteristics of a triode structure CNT electron gun were studied, which is connected in series with a commercial high-voltage metal-oxide-semiconductor field-effect transistor (HV-MOSFET). Because the commercially available HV-MOSFETs are designed for high-power applications, typically with on-currents above milliamperes, there is a mismatch between the operating currents of MOSFETs and cold cathode electron guns. The problem was solved by using the MOSFET in the subthreshold region. Low-voltage high-precision control of emission current was realized, as well as high current stability.

## 2. Experimental

The CNTs were prepared by thermal chemical vapor deposition (TCVD) on a round stainless steel (SS 304) substrate of 1.6 mm in diameter and 0.2 mm in thickness. The details of the preparation process can be found in previous works [37,39]. Briefly, the 15 nm thickness Al_2_O_3_ thin film and 2 nm thickness Fe thin film were first deposited by sputtering acting as the buffer layer and catalyst. Then, the substrate was placed in the center of the quartz tube furnace and the tube furnace was pumped to 1.5 × 10^−3^ Pa. The mixture gasses of hydrogen and argon were continuously introduced while the tube was heated up to 650 °C in 60 min. Following a 30 min hydrogen treatment and a subsequent 10 min temperature increase, CNTs were grown on the substrate by introducing acetylene and hydrogen, when the temperature reached 750 °C. After 20 min, the reaction gas was stopped, and the mixture gas of hydrogen and argon was resumed until the temperature lowered to room temperature. The morphology of the prepared CNTs was characterized by SEM (SUPRA™60, Zeiss, Oberkochen, Germany). The structure of the CNTs was also examined by a Raman spectroscope (FLSP920, Edinburgh Instruments, Livingston, UK).

The structure of the CNT cold cathode electron gun triode is shown in Figure 1. The electron gun is composed of CNT cold cathode and control gate. The gate is a molybdenum mesh with a diameter of 2 mm and is placed above the cathode at a distance (d) of 100 μm. The mesh holes have a diameter of 90 µm and a spacing of 30 µm. The distance (D) between the CNTs cold cathode and anode is approximately 5 mm.

Figure 1 also shows the measurement set-up. The CNT cold cathode electron gun is located in a high vacuum chamber with a base pressure of 1.3 × 10^−5^ Pa and the MOSFET is on a PCB board outside the vacuum chamber. When the switch is set to position I, the characteristics of the CNT cold cathode electron gun can be measured; when the switch is set to position II, the CNT cold cathode electron gun is connected in series with the MOSFET (model: C2M0080120D, CREE, Durham, NC, USA) and the characteristics of the triode electron gun with MOSFET are measured.

The gate of the CNTs cold cathode electron gun is connected to a linear high voltage direct current power supply (HV-HVL 2.5kV-10, Hangyu, Shanghai, China), capable of biasing up to a maximum voltage of 2.5 kV and an individual digital voltage source (Glow 28520, Glow, Qinhuangdao, China) is used to supply the anode voltage. The drain of MOSFET was connected in series with the cathode of the CNT electron gun. The source of MOSFET was grounded through a digital amperemeter. The gate of MOSFET was linked to a regulated power supply (UTP1306S, UNI-T, Dongguan, China), which can provide a highly stable voltage output. By using the above set-up, the relationship of anode current (I_a_), gate current (I_g_), cathode current (I_c_) of triode CNT electron gun, and drain-source current (I_DS_) of MOSFET versus triode gate voltage (V_g_), MOSFET gate voltage (V_G_), and anode voltage (V_a_) can be recorded.

## 3. Results and Discussion

### 3.1. Characterization of the Carbon Nanotube Cathode

Figure 2 shows the morphology of the prepared CNTs. Figure 2a shows the top view of the prepared CNTs and the corresponding magnified SEM image is shown in Figure 2b. The prepared CNTs are disordered and densely packed, with diameters less than 100 nm and lengths ranging from a few micro-meters to several tens of micro-meters. Figure 2c,d are the sideview of prepared CNTs. The CNTs protrude from the surface of the sample which could act as emitters for field emission. We notice most nanotubes are curved and during field emission, these carbon nanotubes will be stretched, and “head-shaking” effect is inevitable as reported in the literature [40]. All these could lead to the current instability in field emission current. Figure 2e shows the Raman spectrum. The Raman shift at 1344 cm^−1^ and 1586 cm^−1^ were observed, corresponding to the D peak and G peak of CNTs. The intensity ratio of I_G_/I_D_ is about 0.79, indicating a relatively high defect level in the graphene structure of prepared CNTs [41,42]. Some early studies have indicated that defects in CNTs could enhance the field emission [43,44,45].

### 3.2. Characteristics of Triode CNT Cold Cathode Electron Gun

The cathode current versus gate voltage characteristics of the CNT cold cathode electron gun without MOSFET were measured under different anode voltages, as shown in Figure 3a. As the anode voltage increased from 750 V to 1500 V, the cathode current rose from 78.2 μA to 104.5 μA (@ Vg = 701 V), representing an approximate increase of 34%. This indicated that while the cathode emission is primarily regulated by the gate voltage, the electric field induced by the anode voltage at the cathode surface also had a significant influence on the cathode current. The F-N curve for the cathode current versus the gate voltage was plotted (the inset of Figure 3a), which was observed to be close to linearity. The field enhancement factor of about 6460 can be calculated from this linear relationship under different anode voltages [46].

Figure 3b shows the anode current (I_a_), gate current (I_g_), and cathode current (I_c_) versus gate voltage (V_g_) characteristics of the electron gun measured under 1500 V anode voltage. The maximum current of 104.5 μA has been achieved at a gate voltage of 701 V, which corresponds to a current density of 5.2 mA/cm^2^ at the cathode (@~7 V/µm).

The electron transmission rate of the gate was calculated using I_a_/I_c_ and the results are shown in the inset of Figure 3b. A SEM picture of the gate mesh used in the electron gun is also shown in the inset of Figure 3b. The transmission rate was usually determined by the aperture ratio of the gate mesh, which was approximately 55% according to our design. The experimentally observed electron transmission rate is in the range from 40% to 60% under different gate voltages. The variation in the transmission rate is due to the instability of the current. The relative position of the CNTs to the gate mesh affected the transmission. We could increase the aperture ratio of the mesh to increase the transmission rate. However, the gate mesh strength will be reduced by increasing the aspect ratio, impacting its stability under high current bombardment. Here, we adopted a trade-off solution to balance these factors. Some researchers employed patterned CNT arrays corresponding to the aperture to improve the transmission rate, but this introduced complexity in the fabrication process [47].

The anode and gate current versus anode voltage characteristics of the CNT electron gun were measured under different gate voltages, as shown in Figure 4. When the device operates normally, the anode voltage should be larger than the gate voltage. It can be seen when the anode voltage is large enough, the anode current begins to saturate as shown in Figure 4a. This indicates that all the electrons passing through the gate are collected by the anode when the anode voltage reaches a certain value. The corresponding gate current also remains constant in this region as shown in Figure 4b. The saturation current increases with the gate voltage. When the gate voltage changes from 420 to 700 V, the anode current increases from 0.43 μA to 59.96 μA, and the corresponding gate current changes from 0.38 μA to 67.59 μA.

When the anode voltage was less than the gate voltage, the anode current still existed, which decreased as the anode voltage decreased. This suggested that electrons, after being accelerated by the gate, maintained enough velocity to overcome the decelerating electrical field of the anode and reach the anode. The gate current gradually increases when the anode voltage decreases during this process, as shown in Figure 4b. As the anode voltage continued to decrease to a sufficiently low value, the anode current ceased to be positive, and the gate current reached the maximum. The gate collected all the electrons emitted from the cathode at this region.

### 3.3. I-V Characteristics of Triode CNT Cold Cathode Electron Gun in Series with MOSFET

The transfer and output characteristics of the HV MOSFET were tested before connecting to the CNT cold cathode electron gun, as shown in Figure 5. Figure 5a shows the transfer characteristic and a subthreshold swing of about 3.43 V/decade can be calculated. The threshold voltage (V_th_) can be estimated to be approximately 4.52 V from the inset of Figure 5a [48]. Figure 5b,c show the output characteristics of the MOSFET. The drain currents were, respectively, ~2.42 mA to ~0.78 μA when the MOSFET gate voltage was changed from 5.0 V to 4.0 V in the saturation region. When the MOSFET drain voltage (V_DS_) is relatively low, the MOSFET is in the linear region, where the drain current (I_D_) is linearly related to the drain voltage (V_DS_).

Figure 6 shows the I_a_-V_g_ characteristics of the CNT electron gun in series with MOSFET while the anode voltage was set as 1500 V. The gate voltage of MOSFET can effectively tune the anode current. When the MOSFET gate voltage (V_G_) is larger than 4.64 V, the I_a_-V_G_ characteristics are the same as the electron gun without MOSFET, because the on-current of a MOSFET exceeds that of an electron gun, and it operates without current limitations. When the MOSFET gate voltage (V_G_) is smaller than 4.64 V, i.e., the MOSFET is in the subthreshold region, we observed the anode current became controllable by the MOSFET. As the MOSFET gate voltage decreases from 4.6 V to 4.1 V in steps of 0.02 V in the subthreshold region, the anode current decreases from 40 µA to 1 µA, as shown in Figure 6b. The results show that the anode current of the electron gun can be accurately regulated by precisely controlling the MOSFET gate voltage in the subthreshold region.

Figure 7 shows the relationship between anode voltage and anode current under the MOSFET gate voltage (V_G_) of 4.6 V and 4.4 V at the same scale. For V_G_ = 4.6 V, when the gate voltage of the CNT electron gun is relatively low (V_g_ < 700 V), the corresponding cathode current is less than the on-currents of MOSFET, resulting in no current limitation. The relationship between anode voltage and anode current (Figure 7a) is the same as that without MOSFET. When the MOSFET gate voltage is 4.4 V, the cathode current is limited by the MOSFET. The corresponding anode saturates when increasing gate voltage (V_g_) to a specific value. Therefore, we observed from Figure 7b that the anode current is limited to approximately 9.8 μA when the V_g_ reaches the saturation region.

### 3.4. Current Stability of Triode CNT Cold Cathode Electron Gun in Series with MOSFET

Finally, the current stability was tested for the CNT electron gun without and with the MOSFET, as shown in Figure 8. The current fluctuation φ was calculated by the following [49]:(1)$$\varphi =\sum \begin{array}{c} n\\ i=1 \end{array}\frac{\left|I_{i}-I_{average}\right|}{n.I_{average}},$$
where *I_i_* is the cathode current at each moment and *I_average_* is the average current. For the CNT electron gun without MOSFET, the calculated cathode current fluctuation was 7.9% and the corresponding anode current fluctuation was 8.6%. For the CNT electron gun in series with the MOSFET, the calculated cathode current fluctuation was 0.87% and the corresponding anode current fluctuation was 2.3%. Compared to the CNT electron gun without MOSFET, the stability of the cathode current and anode current improved by 89% and 73%, respectively, indicating that the MOSFET has the function of stabilizing the current. However, the anode current fluctuation is still relatively high for some applications. The reason is that the current collected by the gate is not constant during the operation. It might be induced by the deformation of gate mesh under electron bombardment or the change in gate-to-CNT distance due to the head-shaking effect of CNTs. To solve this problem, one can increase the robustness of the gate mesh by optimizing the thickness, aperture ratio, or material of the mesh. An external current control circuit connected with a gate electrode could also help to enhance the stability.

Table 1 compared our results with the characteristics of the CNT electron gun with active device control reported in the literature. From the table, we conclude that active device control can increase the stability of the emission current. In the discrete CNT cold cathode electron gun, usually the cathode is more easily stabilized because the cathode current is totally controlled by the active device, while the anode current is more unstable, because the anode current stability is further influenced by the robustness of the gate. More work is needed to further optimize the gate structure and introducing the gate current control might be a promising approach to be explored.

## 4. Conclusions

In this study, the characteristic of CNT cold cathode electron gun in series with MOSFET was studied. The anode current of the electron gun can be accurately regulated from 40 µA to 1 µA when the MOSFET gate voltage was precisely controlled from 4.6 V to 4.1 V in the subthreshold region. The current stability was studied, and the cathode current fluctuation was 0.87% in 10 h continuous operation, corresponding anode current fluctuation was 2.3%. Our study demonstrated the CNT electron gun can be precisely controlled and high stable current can be achieved by connecting with commercial HV-MOSFETs working at the subthreshold region.

## Figures and Tables

**Figure 1 nanomaterials-14-01260-f001:**
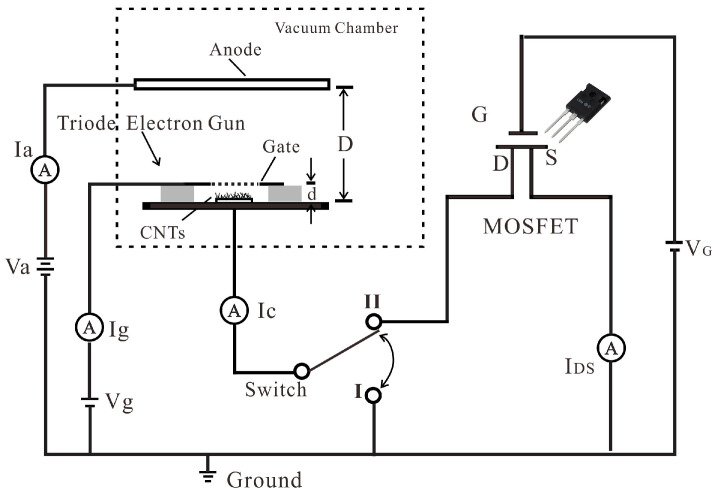
The schematic of the structure and measurement set-up of the CNT cold cathode electron gun in series with MOSFET.

**Figure 2 nanomaterials-14-01260-f002:**
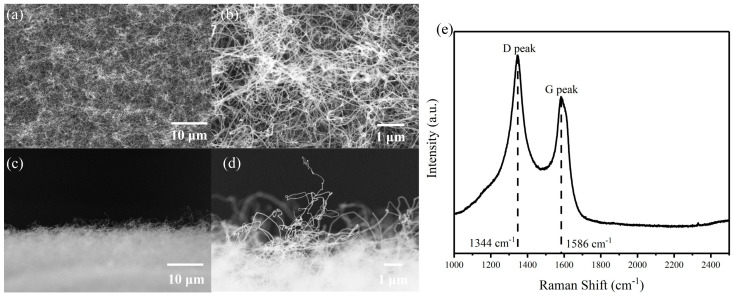
(**a**,**b**) The top view and (**c**,**d**) sideview morphology of the prepared CNTs under different magnifications. (**e**) The Raman spectrum of the prepared CNTs.

**Figure 3 nanomaterials-14-01260-f003:**
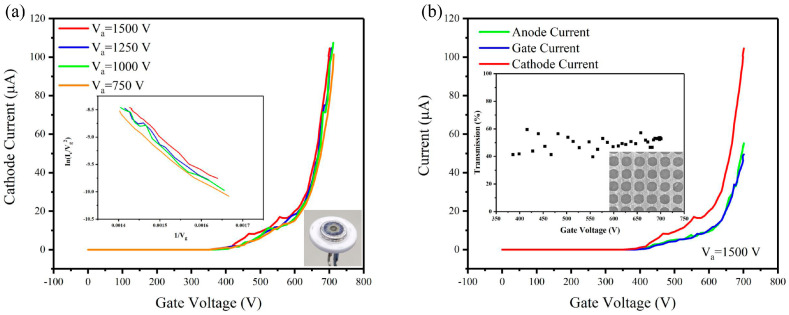
(**a**) I_c_–V_g_ curves under different anode voltages (Inset: corresponding F-N plot) (**b**) I–V curves of the CNT cold cathode electron gun (Inset: transmission rate and the SEM picture of part of the mesh gate).

**Figure 4 nanomaterials-14-01260-f004:**
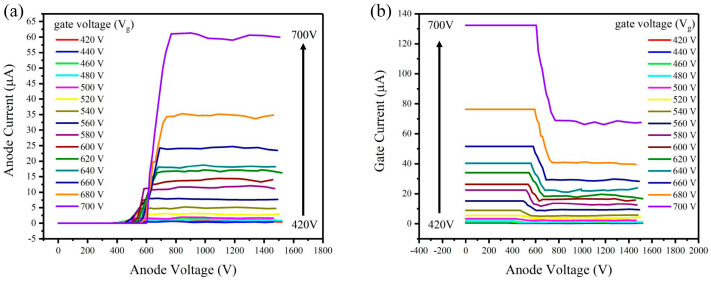
The (**a**) anode current and (**b**) gate current versus anode voltage characteristics of the electron gun under different CNT electron gun gate voltages.

**Figure 5 nanomaterials-14-01260-f005:**
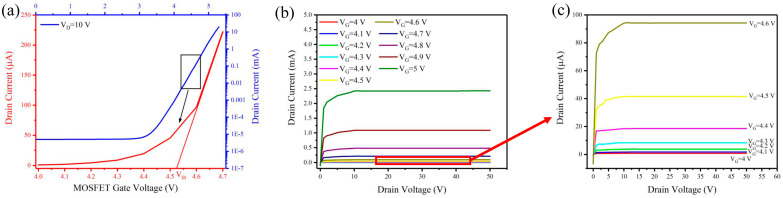
The electrical characteristics of the MOSFET. (**a**) Transfer characteristics; (**b**) output characteristics; (**c**) magnified output characteristics near the subthreshold voltage.

**Figure 6 nanomaterials-14-01260-f006:**
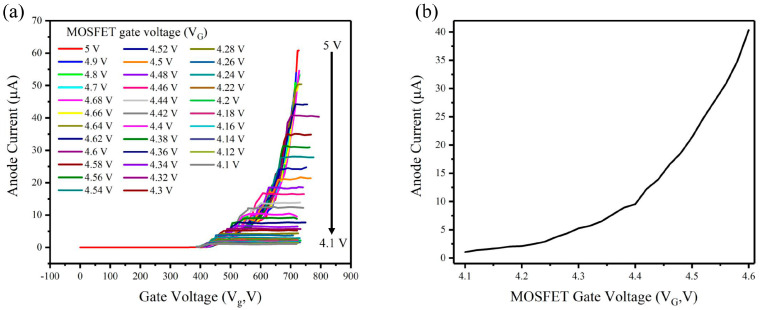
(a)The I_a_-V_g_ characteristics under different MOSFET gate voltages for the CNT cold cathode electron gun in series with MOSFET (b) the relationship between MOSFET gate voltage and anode current.

**Figure 7 nanomaterials-14-01260-f007:**
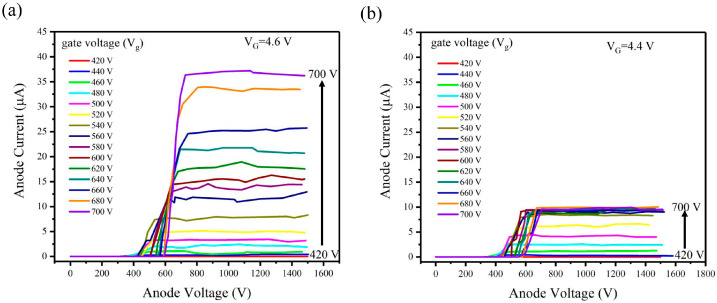
The relationship between anode voltage and anode current in different MOSFET gate voltage (**a**) V_G_ = 4.6 V (**b**) V_G_ = 4.4 V.

**Figure 8 nanomaterials-14-01260-f008:**
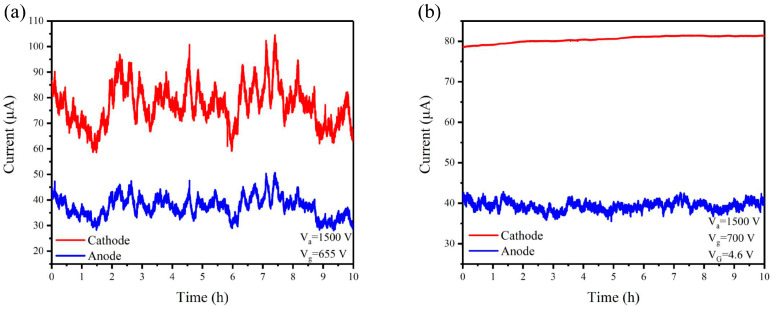
The current stability over 10 h. (**a**) CNT electron gun. (**b**) CNT electron gun in series with MOSFET when the MOSFET gate voltage was 4.6 V.

**Table 1 nanomaterials-14-01260-t001:** The comparison of the characteristics of CNT electron guns with active device controls reported in the literature.

Cathode	Electron Gun Structure	Control Unit	Active Device Connection Method	Operation Mode	Operation Current	Current Fluctuation	Ref.
Mo-tip	Triode	a-Si TFT	Integrated	pulse	~5 µA (cathode)	/	[32]
ZnO NW	Diode	MOSFET	Integrated	direct current	~1.15 µA (cathode)	3.2% over 3 h	[33]
Carbon Nanotubes	Triode	TFT	Integrated	direct current	11 µA (anode)	<2% over 1 h	[34]
Carbon Nanotubes	Triode	IGBT	Discrete	direct current	~50 µA (cathode)	0.22% over ~0.8 h	[36]
Carbon Nanotubes	Triode	Two transistors	Discrete	direct current	1.66 mA (cathode)	0.13% over 2000 s	[38]
Carbon Nanotubes	Triode	Two transistors	Discrete	direct current	1.6 mA (anode)	0.16% over 2000 s	[38]
Carbon Nanotubes	Triode	HV-MOSFET	Discrete	pulse	~10 mA (cathode)	/	[50]
Carbon Nanotubes	Triode	Current controller	Discrete	pulse	~10 mA (cathode)	/	[51]
Carbon Nanotubes	Triode	Two transistors	Discrete	direct current	~2 mA (cathode)	<0.1% over 600 s	[52]
Carbon Nanotubes	Triode	MOSFET	Discrete	direct current	~80 µA (cathode)	0.87% over 10 h	this work
Carbon Nanotubes	Triode	MOSFET	Discrete	direct current	~40 µA (anode)	2.3% over 10 h	this work

## Data Availability

Data are contained within the article.

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
