# Peer review of "Characteristics of Carbon Nanotube Cold Cathode Triode Electron Gun Driven by MOSFET Working at Subthreshold Region"

_nanomaterials, 2024, doi:10.3390/nano14151260_

Round 1

Reviewer 1 Report

Comments and Suggestions for Authors

The research paper entitled “Characteristics of carbon nanotube cold cathode triode electron gun driven by MOSFET working at subthreshold region” discusses the characteristics of carbon nanotube electron gun in series with metal-oxide-semiconductor field-effect transistor. The manuscript can be accepted for publication in Nanomaterials after major revision. Please, find my comments below.

- Add more current references (2023 – 2024).

- What is the novelty of the work? Please, highlight it better.

- Although the information is detailed, the article could benefit from clearer structuring and more coherent wording to facilitate reader understanding. Some sections are too dense and could be broken up into smaller, more clearly defined paragraphs.

- A more extensive discussion of the methodological limitations and challenges encountered during the experiments would be beneficial. It could also include suggestions for future research to address these limitations.

- Quality of CNT images (Fig2) should be improved.  Please, add peculiarities of morphology and its discussion.

Reviewer 2 Report

Comments and Suggestions for Authors

The authors report the characteristics of carbon nanotube (CNT) electron gun in series with metal-oxide-semiconductor field-effect transistor (MOSFET). The anode current of the electron gun was accurately regulated by precise control of the MOSFET gate voltage in the subthreshold region which also greatly improved the current stability.

The manuscript is well organized with relevant data and results. However I recommend the authors to include additional discussion for better understanding of the readers.

1. It is a little confusing whether there can be a strategy to improve the current stability by modifying or optimizing the carbon nanotube cathode itself. Can any type of cathode be used with the MOSFET control unit for stable performance?

2. In section 3.1, the authors state the prepared CNTs are disordered and has a relatively high defect level. Is this intentional? The authors should discuss more in detail about the morphology and its influence on the device stability.

3. In section 3.3 the authors state that a subthreshold swing of about 3.43 V/decade can be calculated for the MOSFET that was employed. Did the authors test with MOSFETS with higher/lower subthreshold swing? I believe this will have a significant effect on the device.

Comments on the Quality of English Language

English must be improved to be accepted for publication. Grammar mistakes can be found throughout the manuscript.

For example, line 39; applied a voltage ->applying voltage

line 70; prepared->prepare

line 189; It can see -> It can be seen

Many more can be found and the authors should revise the manuscript carefully.

Round 2

Reviewer 1 Report

Comments and Suggestions for Authors

I think that now the paper can be accepted for publication.